# Dual-Stage Gradient Projection Based Continual Learning: Enhancing Plasticity and Preserving Stability

## Abstract

In continual learning, gradient projection algorithms avoid forgetting by projecting the gradient onto the orthogonal complement of the feature space of previous tasks, thereby ensuring the model's stability. However, strict orthogonal projection can cause the projected gradient to deviate sharply from the original gradient, damaging the model's learning ability to new tasks and reducing its plasticity. Gradient-projection methods that relax the orthogonality constraint alleviate the deviation introduced by strict projection, yet the degree of gradient distortion remains large and the model's plasticity still needs improvement. To address such an issue, we propose a continual-learning method based on two-stage gradient projection that improves the model's plasticity for new tasks while preserving its stability on previous tasks. Specifically, in the first stage, we design a loss-sensitive space (LSS) regularization term (soft regularization) on top of the cross-entropy loss to constrain the gradient to update as closely as possible along directions orthogonal to the feature space of previous tasks, thereby maintaining plasticity. In the second stage, a scaled projection (hard projection) further constrains the gradient to update along directions approximately orthogonal to the feature space of previous tasks, thus ensuring stability. Experimental results on three benchmark image classification datasets demonstrate that our method, for the first time, reduces the gap between the achieved classification accuracy and the task-specific upper bound (multitask) to within roughly 2%, indicating that the model possesses both strong plasticity and stability.

## 1 Introfction

Continual learning (CL) allows a model to acquire new knowledge without forgetting previously learned information French (1999); McCloskey & Cohen (1989). The capacity to preserve earlier knowledge while studying a new task is called *stability*, and the capacity to absorb new information is called *plasticity*. Balancing these two objectives is known as the stability–plasticity dilemma Abraham & Robins (2005), which is a significant challenge.

Among various CL paradigms, **gradient-projection methods** are attractive for their negligible memory overhead and algorithmic simplicity. Unlike replay approaches that store past data Chaudhry et al. (2019c); Hyder et al. (2022); Prabhu et al. (2020b) or architectural solutions that grow subnetworks dynamically Guo et al. (2020); Mallya & Lazebnik (2018), gradient-projection algorithms leave the original network intact and require no sample rehearsal. They mitigate catastrophic forgetting by constraining the update for a new task to be orthogonal to a space spanned by representations of earlier tasks (e.g., GPM Saha et al. (2021) and OWM Zeng et al. (2019)). Strict orthogonality excels at stability but often harms plasticity, because it suppresses gradient components that are useful for the current task. Recent variants soften this constraint by introducing a scaling matrix that controls how much the gradient is allowed to approach the protected space, for example, SGP Saha & Roy (2023) applies a diagonal scaling on the protected basis to modulate the projected component, and SD Zhao et al. (2023) separates plasticity and stability spaces to improve the overall trade-off.

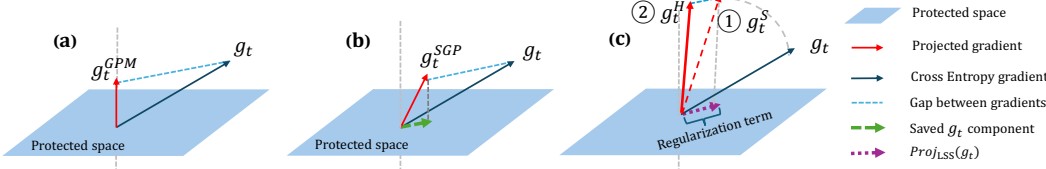

Figure 1: The red arrows denote the gradients actually used to update the parameters after projection. $g_t$ is the original cross-entropy gradient, and the blue dashed lines indicate the discrepancy between gradients. **(a)** $g_t$ is directly projected onto the orthogonal complement of the protected space, yielding $g_t^{\text{GPM}}$. The difference between $g_t$ and $g_t^{\text{GPM}}$ is large, indicating severe distortion. **(b)** $g_t$ is projected onto the orthogonal complement of the scaled space, so part of its components (green arrow) is preserved, producing $g_t^{\text{SGP}}$. The gap between $g_t$ and $g_t^{\text{SGP}}$ is reduced but remains considerable. **(c)** By minimizing the projection of $g_t$ onto the protected space LSS (purple arrow), $g_t$ is refined to $g_t^{\text{S}}$, which lies closer to the orthogonal direction. Projecting $g_t^{\text{S}}$ onto the orthogonal complement then gives $g_t^{\text{H}}$, and the difference between $g_t^{\text{S}}$ and $g_t^{\text{H}}$ becomes much smaller.

However, when the gradient is nearly parallel to the feature space, projecting it onto the orthogonal complement pushes it far from the original gradient, causing severe distortion in both direction and magnitude (Fig. 1 (a)). Methods such as SGP relax the strict constraint to alleviate this, but for critical directions where the constraint cannot be loosened, substantial distortion still occurs (Fig. 1 (b)).

**Our idea**   To address the above issue, we propose a two-stage gradient projection strategy based on a loss-sensitive space (LSS) to reduce the distortion introduced by conventional projection operators and thereby improve plasticity. As shown in Fig. 1 (c), if we first restrict the gradient to update along directions approximately orthogonal to the feature space and then apply the standard projection, the projection no longer induces large angular deviation or significant shrinkage in length. Specifically, in the first update stage (Fig. 1(c)), we augment the cross-entropy loss with a regularization term that minimizes the gradient's component inside the previous-task feature space, forcing the update to move as close as possible to its orthogonal directions. To more accurately quantify the importance of each basis vector in that space, we introduce scaling coefficients derived from the second-order information of past-task losses: Using a diagonal Fisher approximation and the quadratic term of a second-order Taylor expansion, we estimate the loss increase induced by parameter perturbations and use this estimate to rescale the basis vectors. In the second stage, we apply a standard projection to the gradient obtained in stage one. Leveraging SGP's scaled orthonormal basis, we project the gradient onto its orthogonal complement to guarantee stability. Because the gradient has already been guided toward nearly orthogonal directions, this final projection induces only negligible distortion. Our contributions are as follows:

1. We propose a novel insight: by using a loss-based regularization term to constrain the gradient update direction, reducing the distortion caused by projection operators.

2. We propose a two-stage gradient projection strategy combining soft regularization with standard projection, retaining greater plasticity while maintaining stability.

3. To design the soft regularization term, we construct a loss-sensitive space (LSS) from the second-order information of past tasks' losses to quantify each basis vector's importance, and we provide a theoretical justification for its introduction.

4. Experiments on three image-classification benchmarks confirm that our approach retains greater plasticity while preserving stability, resulting in improved performance.

## 2   RELATED WORK

**Non-Projection Continual-Learning Methods**   Continual learning methods are commonly categorized as replay-based, regularization-based, architectural-based, and optimization-based approaches Wang et al. (2024). Replay-based methods usually retain a small buffer of past samples and interleave them with new data, such as **GDumb** retrains a model from scratch on the buffered set, whereas **A-GEM** samples that buffer online to bound interference with earlier tasks Prabhu

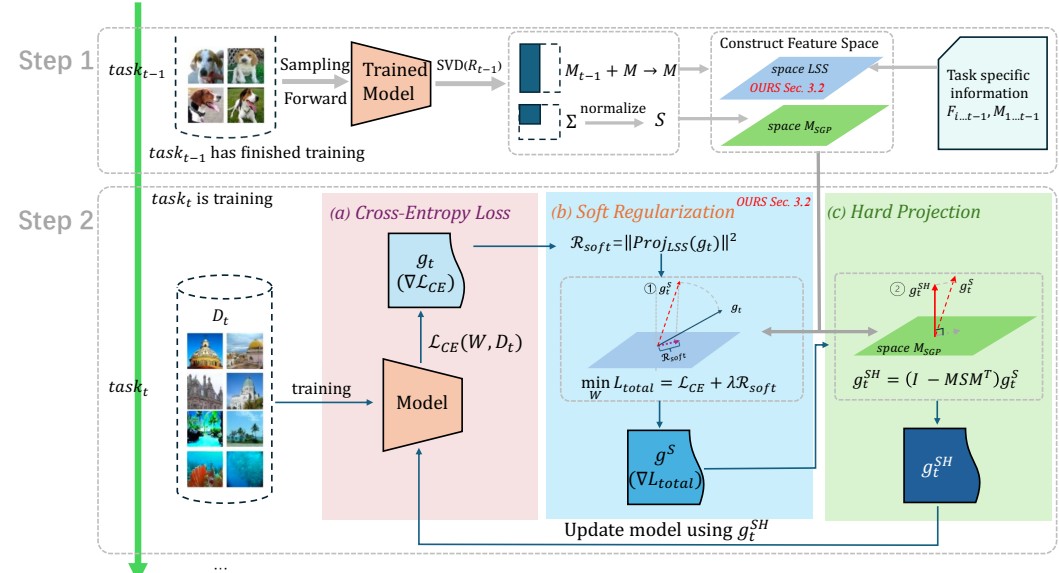

Figure 2: **This figure outlines our pipeline**. Step 1 is the feature-space construction phase of projection-based methods: We sample activations and perform an SVD to derive the task-specific subspace $M_{t-1}$, update the global feature space $M$, and then combine $M_{t-1}$ with past task information to construct two scaling spaces. Step 2 is the actual training loop. After the current-task data pass through the network, stage (a) computes the gradient of $\mathcal{L}_{ce}$, denoted $g_t$; stage (b) combines $g_t$ with the LSS to create a soft regulariser that drives the gradient toward directions orthogonal to the LSS; stage (c) applies the conventional hard projection to the resulting gradient $g_t^S$. Because stage (b) has already pushed the update toward the orthogonal complement, the subsequent hard projection removes far fewer components, preserving plasticity while still protecting prior knowledge. The two key contributions of this paper appear in *Step 1*, where we construct a novel loss-sensitive scaling (LSS) space, and in *Step 2 (b)*, where the LSS is used to build a soft regulariser that is optimised jointly with the cross-entropy loss $\mathcal{L}_{ce}$.

et al. (2020a); Lopez-Paz & Ranzato (2017). Regularization techniques constrain parameter updates to stay close to previously important values; a seminal example is Elastic Weight Consolidation (**EWC**), which adds a Fisher-information penalty, while Learning without Forgetting distils knowledge through soft targets without keeping old samples Kirkpatrick et al. (2017); Li & Hoiem (2017). Architectural solutions dynamically allocate or recycle capacity, such as **PackNet** iteratively prunes and re-grows task-specific subnetworks, and **Tinysubnets** combines layer-wise adaptive pruning, quantization, and weight sharing to exploit sparsity and delay capacity saturation while maintaining competitive accuracy Mallya & Lazebnik (2018); Pietron et al. (2025).

**Gradient-Projection Methods** Optimization based methods adjust the learning dynamics themselves—e.g., by adapting gradient directions to reduce interference between tasks and improve overall performance. **OWM** Zeng et al. (2019) constructs a projection operator via recursive least squares but still shows noticeable forgetting over long task sequences. **Adam-NSCL** Wang et al. (2021) projects gradients onto the null space of the feature-covariance matrix. **Gradient Projection Memory (GPM)** Saha et al. (2021) samples layer-wise activations, applies SVD, and projects gradients onto the orthogonal complement of a low-rank subspace. **Class Gradient Projection (CGP)** Chen et al. (2022) replaces task-level subspaces with class-level ones. **TRGP** Lin et al. (2022b) rescales prior parameters near the current task and then performs orthogonal projection; **CUBER** Lin et al. (2022a) selects gradients beneficial to past tasks by measuring similarity between new and old gradients. **SGP** Saha & Roy (2023) tilts gradients toward low-energy directions; **SD** Zhao et al. (2023) decouples plasticity and stability spaces; and **GPCNS** Yang et al. (2024) builds a joint gradient–feature space to enhance plasticity. Above methods do not constrain update directions to reduce projection-induced distortion. To our knowledge, this is the first work to address that projection distortion using a two-stage projection approach.

## 3 METHOD

In this section, we first review the preliminary, including the feature-basis construction step and the model update step. We then introduce our two-stage gradient-projection strategy, which optimizes update directions to prevent forgetting. In Step 1 (Fig. 2, Step 1. space LSS), we construct a loss-sensitive space (LSS) to avoid the strict constraints. In Step 2 (Fig. 2, step 2 (a)–(b)), we add a regularizer to constrain the gradient direction and jointly optimize it with the cross-entropy loss to preserve plasticity as the first stage. We then apply the standard projection operator to project the total loss gradient $g_t^S$ onto the orthogonal complement of the protected feature space as the second stage (Fig. 2, step 2 (c)), ensuring stability with minimal additional distortion.

### 3.1 PRELIMINARY

**Continual Learning Setting**  In continual learning, a neural network $f$ parameterised by $\mathbf{W} = \{\theta^\ell\}_{\ell=1}^L$ is trained sequentially on a stream of tasks $\mathcal{T} = \{t\}_{t=1}^T$. Each task $t$ comes with a dataset $\mathbb{D}_t = \{(x_{t,i}, y_{t,i})\}_{i=1}^{n_t}$ of size $n_t$, where $x_{t,i}$ denotes the input and $y_{t,i}$ its label. After finishing task $t$, the model is parameterised by $\mathbf{W}_t = \{\theta_t^\ell\}_{\ell=1}^L$. The feature produced by layer $\ell$ is written $x_{t,i}^\ell$, with $x_{t,i}^1 = x_{t,i}$. The training loss for task $t$ is denote as $\mathcal{L}_t = \mathcal{L}_t(\mathbf{W}, \mathbb{D}_t)$.

Let $R_{t-1}^\ell = \left[x_{t-1,1}^\ell, \ x_{t-1,2}^\ell, \ \ldots, \ x_{t-1,n_s}^\ell\right]$, denote the representations sampled from the $t-1$-th task at layer $\ell$, and let $\Delta\theta_{t-1}^\ell$ be the parameter change induced by learning the $t$-th task. When learning a new task t, the parameter tensor will deviates from its optimal value for former tasks due to the update $\Delta\theta_{t-1}^\ell$. This process can be formally described as $\theta_t^\ell R_{t-1}^\ell = \left(\theta_{t-1}^\ell + \Delta\theta_{t-1}^\ell\right)R_{t-1}^\ell = \theta_{t-1}^\ell R_{t-1}^\ell + \Delta\theta_{t-1}^\ell R_{t-1}^\ell$. The $\theta_t$ will keep the knowledge of task $t-1$ if $\theta_t^\ell R_{t-1}^\ell = \theta_{t-1}^\ell R_{t-1}^\ell$. That means if $\Delta\theta_{t-1}^\ell R_{t-1}^\ell = 0$ is satisfied, the forgetting issue will be overcome, which motivates the gradient-projection method described below (see also Fig. 2, step 1 and step 2(a)/2(c)).

**Step 1: Construct Feature Bases After Completing Task** $t-1$**.**  After finishing the training of task $t-1$, we extract each layer's representations $R_{t-1}^\ell$ and define its specific feature space as $M_{t-1}^\ell = \text{span}\left\{R_{t-1}^\ell[1:k]\right\}$, where the $R_{t-1}^\ell = U_{t-1}^\ell \Sigma_{t-1}^\ell V_{t-1}^{\ell\top}$ is computed with SVD and $k$ is the smallest $k$ s.t. $\|\Sigma_{1:k}\|_F^2 \geq \epsilon\|\Sigma\|_F^2 (\epsilon \in [0,1]$ is the threshold). Let $\bar{M}^\ell(t-2)$ denote the accumulated space of all tasks up to $t-2$. The updated space after task $t-1$ is

$$\bar{M}^\ell(t-1) = \bar{M}^\ell(t-2) + M_{t-1}^\ell \tag{1}$$

For scale-based methods, one additional procedure computes a scaling diagonal matrix; for example, SGP normalizes the singular values in $\Sigma^\ell$ to obtain the scaling factors $S$.

**Step 2: Update the Model for Task** $t$  When training task $t$, we first compute the cross-entropy gradient $g_t = \nabla_{W^\ell} \mathcal{L}_{\text{CE}}^{(t)}(W, \mathcal{D}_t)$. To curb catastrophic forgetting, we then project $g_t$ onto the orthogonal complement of the accumulated subspace $\bar{M}^\ell(t-1)$, while controlling the degree of orthogonality with a diagonal scaling matrix $S = \text{diag}(s_1, \ldots, s_k)$:

$$g_t^{\text{proj}} = \left(I - \bar{M}^\ell(t-1)\, S\, \bar{M}^\ell(t-1)^\top\right) g_t. \tag{2}$$

For convenience, we introduce the notation $\text{Proj}_M^S(g_t) = \left(M\,S\,M^\top\right)g_t$, $\text{Proj}_{M^\perp}^S(g_t) = \left(I - M\,S\,M^\top\right)g_t$, so that setting $S = I$ recovers the standard unscaled projections onto $M$ and its orthogonal complement.

The method above avoids forgetting by projecting $g_t$ onto the orthogonal complement of $\bar{M}$. This implies that the closer $g_t$ lies to $\bar{M}$ of task $t-1$, the greater the distortion in $\text{Proj}_{\bar{M}}(g_t)$. We observe that if $g_t$ is already orthogonal to $\bar{M}$ before projection, then the discrepancy between $\text{Proj}(g_t)$ and $g_t$ is minimized, causing less harm to plasticity.

### 3.2 DUAL-STAGE GRADIENT PROJECTION

Based on above insight, we propose a two-stage gradient-projection strategy that optimizes update directions to enhance plasticity while preserving stability.

**Soft Projection Stage**  To constrain the gradient update direction, we introduce a regularization term that is jointly optimized with the cross-entropy loss, ensuring a proper balance between task performance and directional restriction (Fig. 2, Step 2(a)–(b)).

During the training of task $t$, we denote the cross-entropy loss function gradient of $\ell$ layer as $g_t^\ell = \nabla_{w^\ell}\mathcal{L}_{\text{CE}}(W, \mathbb{D}_t)$. We then construct a *soft* penalty for $\ell$-th layer

$$\mathcal{R}_{\text{soft}}^\ell = \left\|\text{Proj}_{\bar{M}^\ell(t-1)}^{S_1^\ell}(g_t^\ell)\right\|_2^2 = \left\|\bar{M}^\ell(t-1)S_1^\ell\bar{M}^\ell(t-1)^\top g_t^\ell\right\|_2^2 \tag{3}$$

where $S_1^\ell$ is a scaling matrix attached to $M^\ell(t-1)$. The total loss becomes

$$\mathcal{L}_{\text{total}} = \mathcal{L}_{\text{CE}} + \lambda \sum\nolimits_{\ell=1}^L \mathcal{R}_{\text{soft}}^\ell, \tag{4}$$

However, using a standard orthonormal basis in the regularizer can be overly restrictive, since we only require the gradient to be orthogonal to the most important directions in $M$. To align this constraint with the cross-entropy loss, we replace the original scaling parameter $S_1$ in Eq. equation 3 with a loss-sensitive scaling coefficient (LSS) derived from the curvature information of the previous-task loss.

**Loss-Sensitivity Theoretical Analysis via Taylor Expansion**  Here, we provide a theoretical analysis of the perturbation $\Delta\mathcal{L}_{t-1}$ to the previous-task loss $\mathcal{L}_{t-1}$ caused by the parameter update $\Delta\theta_{t-1}$ during training on task $t$, and decouple this perturbation onto each basis vector of the feature space to characterize the loss change induced by updating along each direction.

During the training of task $t$, let $\mathcal{L}_{t-1}(\theta)$ denote the loss of task $t-1$. Applying the update $\Delta\theta_{t-1}$ from task $t$ perturbs this loss to $\mathcal{L}_{t-1}(\theta_{t-1} + \Delta\theta_{t-1})$. To compute the resulting change in $\mathcal{L}_{t-1}$, we perform a second-order Taylor expansion around the converged parameter $\theta_{t-1}$:

$$\mathcal{L}_{t-1}(\theta_{t-1} + \Delta\theta_{t-1}; D_t) \approx \mathcal{L}_{t-1}(\theta_{t-1}) + \nabla_\theta\mathcal{L}_{t-1}(\theta_{t-1})^\top\Delta\theta_{t-1} + \frac{1}{2}\Delta\theta_{t-1}^\top H_{t-1}\Delta\theta_{t-1},$$

where $H_{t-1} = \nabla_\theta^2\mathcal{L}_{t-1}(\theta_{t-1})$ is the Hessian of task $t-1$ at convergence. Since task $t-1$ has converged, $\nabla_\theta\mathcal{L}_{t-1}(\theta_{t-1}) \approx \mathbf{0}$, the resulting loss change simplifies to $\Delta\mathcal{L} \approx \frac{1}{2}\Delta\theta_t^\top H_{t-1}\Delta\theta_t$. Inspired by EWC Kirkpatrick et al. (2017), we approximate the Hessian $H_{t-1}$ by the diagonal Fisher information matrix $F_{t-1}$: $H_{t-1} \approx F_{t-1}$.

From TRGP theory Lin et al. (2022a;b), during the training of task $t-1$, the parameter update $\Delta\theta_{t-1}^\ell$ lies entirely in the task-specific subspace $M_{t-1}^\ell = \text{span}\{R_{t-1}^\ell\}$. Hence, only perturbations *within* $M_{t-1}^\ell$ can affect $\theta_{t-1}^\ell$. Thus, during training task $t$, the loss change of prior task $t-1$ caused by the perturbation is

$$\Delta\mathcal{L}_{t-1} \approx Proj_{M_{t-1}^\ell}(\Delta\theta_{t-1}^\ell)^\top F_{t-1} Proj_{M_{t-1}^\ell}(\Delta\theta_{t-1}^\ell),$$

where $F_{t-1}$ is the Fisher information matrix (diagonal) for loss function of task $t-1$. Since the feature space is typically scaled by a parameter matrix $S$ to enhance the plasticity of the projected gradient, we next derive the relationship between the perturbation $\Delta\mathcal{L}_{t-1}$ in the unscaled space and its counterpart in the $S$-scaled space.

**Theorem 3.1.** *During the training of task $T_t$, for any layer $\ell$, let $\bar{M}(t-1)$ be the total feature space up to task $t-1$, $S$ the scaling matrix, and $g_t^\ell$ the cross-entropy gradient. For any previous task $j$ with loss $\mathcal{L}_j(\theta_j)$, let $\mathcal{M}_j$ be its feature subspace. The change in $\mathcal{L}_j(\theta_j)$ caused by updating $\theta_t$ with $g_t$ is*

$$\Delta\mathcal{L}_j = \sum_\ell p_\ell^2 \left(Proj_{\mathcal{M}_j}(g_t^\ell)\right)^\top F_j Proj_{\mathcal{M}_j}(g_t^\ell), \tag{5}$$

*where $p_\ell = f(\bar{M}(t-1), S)(g_t^\ell)$, Here $f$ takes a subspace $M$ and a scaling matrix $S$ and returns a linear operator on any gradient $g$.(See proof in supplementary materials)*

Theorem 1 motivates us to scale the space by the change in old-task losses. From the perspective of the variation in the old-task loss, the scaling matrix $S$ accounts for only one term $p_\ell^2$ that contributes to the change of $\mathcal{L}_j$ and ignores the loss's second-order curvature information. Next, we therefore construct the loss-sensitive space.

---

**Algorithm 1** Dual-Stage Gradient Projection

---

**Require:** Task stream $\mathcal{T} = \{\mathcal{D}_1, \ldots, \mathcal{D}_T\}$; network $f_W$; learning rate $\eta$; scale coefficient $\alpha$; soft weight $\lambda$; threshold $\epsilon$ for selecting top-$k$ principal components.

**Ensure:** Trained weights $W_T = \{\theta^\ell\}_{\ell=1}^L$

1: $\bar{M}^\ell(0) \leftarrow \varnothing$ {protected basis of each layer $\ell$}
2: $S_{lss}^\ell, S_{sgp}^\ell \leftarrow I$ {scaling matrix of each layer $\ell$}
3: **for** $t = 1$ **to** $T$ **do**
4:    // Training loop (Fig 2 Step 2)
5:    **while** not converged on $\mathcal{D}_t$ **do**
6:       Sample minibatch $B_t \subset \mathcal{D}_t$
7:       $g_t \leftarrow \nabla_W \mathcal{L}_{\text{CE}}(B_t; W)$
8:       $\mathcal{L}_{\text{total}} \leftarrow \mathcal{L}_{\text{CE}}(B_t; W) + \sum_\ell \|\text{Proj}_{\bar{M}^\ell}^{S_{lss}^\ell}(g_t)\|_2^2$            $\triangleright$ *equation 4*
9:       $g_{\text{S}} \leftarrow \nabla_W \mathcal{L}_{\text{total}}$
10:       **for** $\ell = 1$ **to** $L$ **do**
11:          $g_{\text{SH},\ell} \leftarrow \text{Proj}_{\bar{M}^\ell,\perp}^{S_{sgp}^\ell}(g_{\text{S},\ell})$            $\triangleright$ *equation 9*
12:          $\theta^\ell \leftarrow \theta^\ell - \eta\, g_{\text{SH},\ell}$            $\triangleright$ *equation 10*
13:       **end for**
14:    **end while**
15:    //Update protected feature space (Fig 2 Step 1)
16:    **for** $\ell = 1$ **to** $L$ **do**
17:       Sample $n_s$ activations $R_t^\ell$
18:       Compute $M_t^\ell$ via SVD on $R_t^\ell$ and Fisher matrix $F_t^\ell$
19:       $\bar{M}^\ell(t) \leftarrow \bar{M}^\ell(t-1) \cup M_t^\ell$            $\triangleright$ *equation 1*
20:       Compute scaling matrix $S_{sgp}^\ell$ Saha & Roy (2023)
21:       **for all** new basis $u_{i,t}^\ell \in \bar{M}^\ell(t)$ **do**
22:          $\text{LSW}\big(\bar{u}_{i,t}^\ell\big) = \sum_{j=1}^t \Delta\mathcal{L}_j(F_j, u_{i,t}^\ell, M_j^\ell)$            $\triangleright$ *equation 6*
23:       **end for**
24:       Standardize LSW by Eq. equation 16 and get $\bar{S}_{lss}^\ell$ by Eq. equation 17
25:    **end for**
26: **end for**
27: **return** $W$

---

**Constructing the Loss-Sensitive Scaling Space**   Here, we describe the construction of the LSS scaling weights, which is performed during the feature-space construction phase immediately after completing each task and corresponds to Step 1 (LSS space) of the gradient projection paradigm (Fig. 2 Step 1, pipeline in the supplementary materials).

After the training of task $t-1$, to measure the loss sensitivity of each direction in the protected space $\bar{M}^\ell(t-1) = \big[\bar{u}_{1,t-1}^\ell, \bar{u}_{2,t-1}^\ell, \ldots, \bar{u}_{k,t-1}^\ell\big]$, we substitute $g_t^\ell$ with each basis vector $\bar{u}_{i,t-1}^\ell$ in Eq. equation 15. Since $\|\bar{u}_{i,t-1}^\ell\|_2 = 1$, this quantifies the change in task $j$'s loss due to a unit perturbation along $\bar{u}_{i,t-1}^\ell$. Therefore, based on Theorem B.2, we define the loss-sensitive weight across all tasks $j = 1, \ldots, t-1$ as follows:

$$\text{LSW}\big(\bar{u}_{i,t-1}^\ell\big) = \sum_{j=1}^{t-1} \Delta\mathcal{L}_j(F_j, u_{i,t}^\ell, M_j^\ell)$$
$$= \sum_{j=1}^{t-1} \text{Proj}_{M_j^\ell}\big(\bar{u}_{i,t-1}^\ell\big)^\top F_j \text{Proj}_{M_j^\ell}\big(\bar{u}_{i,t-1}^\ell\big). \tag{6}$$

For all weights, $\text{LSW}_{\text{all}} = \{\text{LSW}(\bar{u}_{m,t-1}^\ell)\}_{m=1}^k$, we normalize them Saha & Roy (2023) by

$$s_{i,t-1} = \frac{(1+\alpha)\,\text{LSW}(\bar{u}_{i,t-1}^\ell)}{\alpha\,\text{LSW}(\bar{u}_{i,t-1}^\ell) + \max_m \text{LSW}(\bar{u}_{m,t-1}^\ell)}. \tag{7}$$

Thus, the loss-sensitive scaling matrix is

$$S_{\text{lss}}^\ell = diag\big(s_{1,t-1}, \ldots, s_{k,t-1}\big). \tag{8}$$

**Hard Projection Stage** After computing the loss-sensitive scaling matrix $S_{\text{lss}}$, we replace the original scaling parameter $S_1$ in Eq. (3) with $S_{\text{lss}}$, yielding a new soft-regularization term for each layer $\ell$: $\mathcal{R}_{\text{soft}}^\ell = \left\| \text{Proj}_{M^\ell}^{S_{\text{lss}}}(g_t^\ell) \right\|_2^2$. Since this soft regularizer only refines the gradient direction without fully preventing forgetting, we then compute the updated gradient of Eq. equation 4: $g_t^{\text{S}} = \nabla_{w^\ell} \mathcal{L}_{\text{total}}$, and apply a hard projection to ensure stability (Fig. 2 part (c)):

$$g_t^{\text{SH},\ell} = Proj_{\mathcal{M}^\perp}^{S_{\text{SGP}}}(g_t^{\text{S},\ell}) = \left( I - \mathcal{M} S_{SGP} \mathcal{M}^\top \right) g_t^{\text{S},\ell}, \tag{9}$$

where $\mathcal{M} = \overline{M}(t-1)$, and $S_{\text{SGP}}$ be another scaling matrix following Saha & Roy (2023), constructed from the singular values $\Sigma$ of $\text{SVD}(R_{t-1}^\ell)$. Then, we update the parameters $\theta_{t-1}^\ell$ with learning rate $\eta$:

$$\theta_{t-1}^\ell \leftarrow \theta_{t-1}^\ell - \eta\, g_t^{\text{SH},\ell}. \tag{10}$$

Finally, the framework is presented in Fig. 2 and main steps of our algorithm are summarized in Algorithm 2. A more detailed version of the algorithm can be found in Algorithm 1 of the supplementary material.

## 4 EXPERIMENTS

**Datasets.** To ensure a fair comparison with previous state-of-the-art continual learning methods, we follow the commonly adopted evaluation protocol and select three benchmark image classification datasets. Specifically, we evaluate our method on Split CIFAR-100 Krizhevsky et al. (2009), CIFAR-100 Superclass Yoon et al. (2020) and Split MiniImageNet Vinyals et al. (2016). Split CIFAR-100 contains 60 000 RGB images over 100 classes split into 10 tasks of 10 classes each (500 train / 100 test images per class, $32 \times 32$ resolution). CIFAR-100 Superclass divides the same 100 classes into 20 semantically related superclasses (5 classes each). Split MiniImageNet is a 100-class subset of ImageNet split into 20 tasks of 5 classes each (500 train / 100 test images per class, $84 \times 84$).

**Implementation Details.** For fair comparison, we adopt the same backbones as GPM, TRGP and SGP on each dataset: a 5-layer AlexNet Krizhevsky et al. (2012) on Split CIFAR-100; a LeNet on CIFAR-100 Superclass; and a reduced ResNet-18 He et al. (2016) on Split MiniImageNet. All methods use task-incremental learning with a separate classifier head per task, trained with SGD (momentum 0.9, weight decay $5 \times 10^{-4}$), batch size 64; 200 epochs per task for Split CIFAR-100 and Split MiniImageNet, 50 epochs for CIFAR-100 Superclass.

**Baselines.** To maintain consistency with GPM, TRGP, CGP and SGP, we exclude any method that increases parameters during training Liang & Li (2023). Following SGP Saha & Roy (2023), we compare against OWM Zeng et al. (2019), A-GEM Chaudhry et al. (2019a), Experience Replay with Reservoir sampling (ER_Res) Chaudhry et al. (2019b), Adam-NSCL Wang et al. (2021), GPM Saha et al. (2021), FS-DGPM Deng et al. (2021), CGP Chen et al. (2022), TRGP Lin et al. (2022b), SGP Saha & Roy (2023) and GPCNS Yang et al. (2024). "Multitask" denotes the upper-bound of learning all tasks jointly Hsu et al. (2018).

**Evaluation Metrics.** We employ average accuracy (ACC) and backward transfer (BWT) Lopez-Paz & Ranzato (2017). ACC denotes the average test accuracy across all $T$ tasks, and BWT measures the average decline in test accuracy for previous tasks after learning the current one: $\text{ACC} = \frac{1}{T} \sum_{i=1}^{T} R_{T,i}, \text{BWT} = \frac{1}{T-1} \sum_{i=1}^{T-1} \left( R_{T,i} - R_{i,i} \right)$, where $R_{j,i}$ is the accuracy on task $i$ after learning task $j$ sequentially.

### 4.1 MAIN RESULTS

In this section the main result is showed in Table 1. We denote any feature space used as the soft constraint by the superscript $S$; for example, $\text{LSS}^S$ indicates that the LSS space is employed in the soft step. Spaces applied in the hard projection are marked with the superscript $H$, e.g. $\text{SGP}^H$ denotes that the SGP scaling space is used for hard projection.

Table 1 shows average accuracy (**ACC**) and backward transfer (**BWT**) for our method ($\text{LSS}^S$ + $\text{SGP}^H$, $\text{LSS}^S$ + $\text{TRGP}^H$) and existing baselines on three benchmarks. On *Split* CIFAR *100*,

Table 1: Comparison results on datasets. We report ACC and BWT over 10 runs with random seeds.

| Method | Split CIFAR-100 | | CIFAR-100 Superclass | | Split MiniImageNet | |
|---|---|---|---|---|---|---|
| | ACC (%) | BWT (%) | ACC (%) | BWT (%) | ACC (%) | BWT (%) |
| Multitask | $79.58 \pm 0.54$ | – | $61.00 \pm 0.20$ | – | $69.46 \pm 0.62$ | – |
| OWM | $50.94 \pm 0.60$ | $-30 \pm 1$ | – | – | $47.48 \pm 1.28$ | $-12 \pm 3$ |
| A-GEM | $63.98 \pm 1.22$ | $-15 \pm 2$ | $50.35 \pm 0.80$ | $-9.5 \pm 0.9$ | $57.24 \pm 0.72$ | $-12 \pm 1$ |
| ER_Res | $71.73 \pm 0.63$ | $-6 \pm 1$ | $53.30 \pm 0.70$ | $-3.4 \pm 0.8$ | $58.94 \pm 0.85$ | $-7 \pm 1$ |
| Adam-NSCL | $73.77 \pm 0.50$ | $-1.6 \pm 0.51$ | $56.32 \pm 0.88$ | $-2.42 \pm 0.93$ | $59.07 \pm 1.10$ | $-4.9 \pm 1.32$ |
| GPM | $72.48 \pm 0.40$ | $-0.9 \pm 0.0$ | $57.72 \pm 0.70$ | $-1.2 \pm 0.4$ | $60.41 \pm 0.61$ | $-0.7 \pm 0.4$ |
| FS-DGPM | $74.33 \pm 0.31$ | $-2.71 \pm 0.17$ | $58.81 \pm 0.34$ | $-2.97 \pm 0.35$ | $61.03 \pm 1.08$ | $-1.96 \pm 0.78$ |
| CGP | $74.26 \pm 0.38$ | $-1.48 \pm 0.78$ | $57.53 \pm 0.52$ | $-1.63 \pm 0.49$ | $60.82 \pm 0.55$ | $-0.33 \pm 0.21$ |
| GPCNS | $74.40 \pm 0.42$ | $-2.16 \pm 0.92$ | $58.50 \pm 0.43$ | $-1.86 \pm 0.83$ | $63.78 \pm 0.62$ | $-2.84 \pm 1.15$ |
| GPM + GPCNS | $73.84 \pm 0.29$ | $-0.26 \pm 0.09$ | $58.19 \pm 0.38$ | $-0.47 \pm 0.34$ | $61.26 \pm 0.44$ | $-1.25 \pm 0.36$ |
| TRGP + GPCNS | $75.58 \pm 0.36$ | $-0.06 \pm 0.33$ | $59.51 \pm 0.32$ | $-0.55 \pm 0.27$ | $66.07 \pm 0.47$ | $0.03 \pm 0.29$ |
| SGP + GPCNS | $76.25 \pm 0.38$ | $-0.13 \pm 0.05$ | $59.14 \pm 0.40$ | $-0.74 \pm 0.36$ | $63.98 \pm 0.53$ | $-0.81 \pm 0.31$ |
| TRGP | $74.46 \pm 0.32$ | $-0.9 \pm 0.01$ | $58.25 \pm 0.21$ | $-1.71 \pm 0.52$ | $61.78 \pm 0.60$ | $-0.5 \pm 0.6$ |
| SGP | $76.05 \pm 0.43$ | $-1.23 \pm 0.75$ | $59.05 \pm 0.21$ | $-1.4 \pm 0.51$ | $62.83 \pm 0.33$ | $-1.12 \pm 0.98$ |
| $LSS^S$+$TRGP^H$ | $\mathbf{78.05 \pm 0.44}$ | $-0.47 \pm 0.01$ | $59.32 \pm 0.05$ | $-1.28 \pm 0.05$ | $66.03 \pm 0.93$ | $-0.62 \pm 0.04$ |
| $LSS^S$ + $SGP^H$ | $76.62 \pm 0.09$ | $-1.22 \pm 0.05$ | $\mathbf{59.51 \pm 0.06}$ | $-1.76 \pm 0.03$ | $\mathbf{67.45 \pm 0.75}$ | $-0.10 \pm 0.83$ |

Table 2: Ablation Study on LSS and the Soft Regularization Term

| Method | SOFT | LSS | CIFAR-100 | | Superclass | | MiniImageNet | |
|---|---|---|---|---|---|---|---|---|
| | | | ACC (%) | BWT (%) | ACC (%) | BWT (%) | ACC (%) | BWT (%) |
| ① $SGP^S$ + $SGP^H$ | ✓ | | $76.28 \pm 0.07$ | $-1.01 \pm 0.04$ | $59.04 \pm 0.03$ | $-2.30 \pm 0.10$ | $66.72 \pm 0.60$ | $-0.71 \pm 0.42$ |
| ② $LSS^S$+$SGP^H$ | ✓ | ✓ | $76.62 \pm 0.09$ | $-1.22 \pm 0.05$ | $59.51 \pm 0.06$ | $-1.76 \pm 0.03$ | $67.45 \pm 0.75$ | $-0.10 \pm 0.83$ |
| ③ $LSS^S$+$LSS^H$ | ✓ | ✓ | $75.12 \pm 0.09$ | $-0.23 \pm 0.06$ | $58.01 \pm 0.03$ | $-1.74 \pm 0.06$ | $65.91 \pm 0.62$ | $0.30 \pm 0.23$ |
| ④ SGP | | | $76.05 \pm 0.43$ | $-1.23 \pm 0.75$ | $59.05 \pm 0.21$ | $-1.4 \pm 0.51$ | $62.83 \pm 0.33$ | $-1.12 \pm 0.98$ |

ours method achieves the best ACC of **78.05%**, surpassing the strongest gradient–projection rival **SGP** by **2.03%**, **TRGP** by **3.59%**, and **GPM** by **5.57%**. Its forgetting remains competitive (BWT $= -1.22\%$), confirming that the additional plasticity induced by the *soft* constraint does *not* compromise stability. On *CIFAR 100 Superclass*, With an ACC of **59.51%**, LSS outperforms TRGP, SGP and GPM by **1.3%**, **0.5%** and **1.8%**, respectively, while keeping BWT at $-1.76\%$. On *Split MiniImageNet*. On the more demanding 20-task stream, LSS lifts ACC to **67.45%**, a gain of **4.6%** over SGP, **5.6%** over TRGP and more than **6%** over GPM, accompanied by the lowest forgetting (BWT $= -0.30\%$).

## 4.2 ABLATION STUDY

In this section, we perform ablation experiments to validate the effectiveness of the Soft–Hard framework and the Loss-Sensitive Space (LSS), as summarized in Table 2.

Adding only the soft step ($SGP^S$+$SGP^H$) increases CIFAR-100 ACC from 76.05% to 76.28% and MiniImageNet ACC from 62.83% to 66.72%, confirming a plasticity gain. Replacing the soft subspace with LSS ($LSS^S$+$SGP^H$) further boosts ACC (e.g. +0.34 on CIFAR-100, +0.73 on MiniImageNet) and reduces BWT, validating LSS. Using LSS for the hard step ($LSS^S$+$LSS^H$) lowers ACC but sharply improves BWT (CIFAR-100 BWT -1.22%→-0.23%), demonstrating that SGP's null-space is key for plasticity while LSS-based projection enhances stability.

## 4.3 PLASTICITY AND STABILITY ANALYSIS

In this section, we analyze the plasticity and stability of the combined $LSS^S + SGP^H$ method (abbreviated as LSS), and study the effect of adding the soft-constraint term to the cross-entropy loss on model plasticity (see supplementary materials for more results).

The first row of Fig. 3b shows the first-pass accuracy of LSS and SGP on each task, reflecting the model's *plasticity*. The second and third rows of Fig. 3b report the post-training accuracy on each task and the corresponding backward transfer (BWT) relative to the first-pass accuracy, illustrating the model's *stability*. Fig. 3a (Left) compares vanilla multi-task learning (MTL) with MTL+SOFT, and plots the corresponding first-pass task accuracies. We see that adding the regularization term has

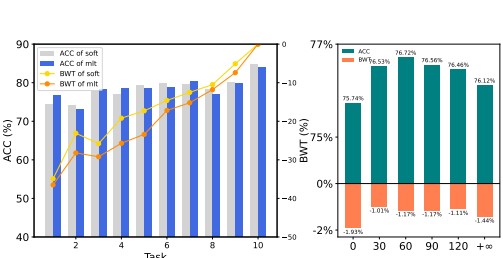

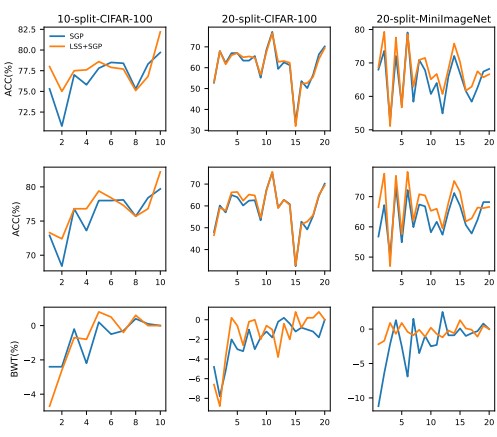

(a) Left: Comparison of the first-task accuracy and per-task BWT between soft constraint and unconstrained MLT on the CIFAR-100 dataset; Right: Impact of scaling parameters $\alpha$ for different LSS bases on accuracy and BWT (Experiments are all conducted on CIFAR-100 dataset).

(b) Comparison of ACC(1st), ACC(last) and BWT for SGP vs. LSS+SGP(OURS).

Figure 3: Overall comparison of methods on CIFAR-100.

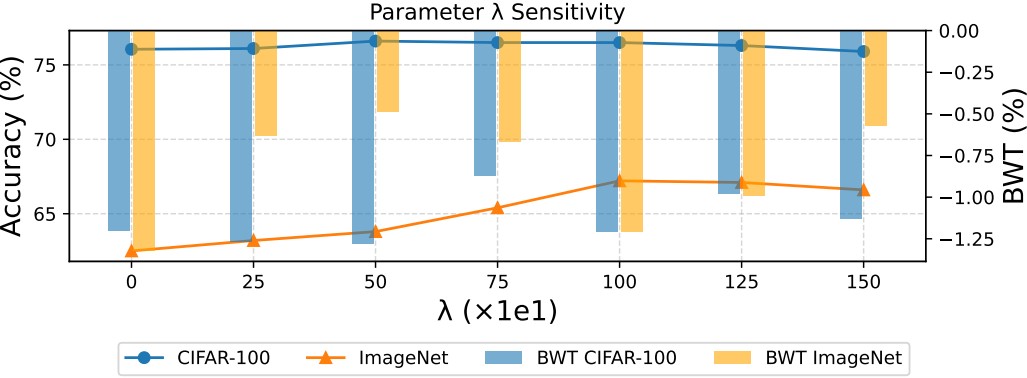

Figure 4: Parameter Sensitivity Analysis

almost no adverse effect on the model's plasticity for new tasks; in fact, it even slightly improves BWT. Fig. 3a (Right) shows the trends of accuracy and BWT on the CIFAR-100 dataset as the soft-weighting parameter $\alpha$ in Eq. equation 16 increases (where "$+\infty$" indicates that the scaling parameter tends to infinity, making the LSS equivalent to an unscaled orthonormal basis).

### 4.4 PARAMETER SENSITIVITY ANALYSIS

We sweep soft constrain parameter $\lambda$ of Eq. equation 4 from 0 to 150 on both **CIFAR-100** and **MiniImageNet**, recording the resulting accuracy and backward transfer (BWT), and the curves are shown in Fig. 4. On CIFAR-100, accuracy rises slowly as $\lambda$ increases, whereas BWT quickly stabilises, suggesting that larger $\lambda$ does not jeopardise stability. On MiniImageNet, small values of $\lambda$ yield low accuracy and BWT, but both metrics improve gradually with larger $\lambda$, indicating that the soft regulariser supplies additional plasticity and stability. Altogether, these observations show that LSS is generally insensitive to the exact value of $\lambda$, while larger value consistently award the model with greater plasticity.

## 5 CONCLUSION

In this work, we have identified and analyzed the factors in gradient-projection operators that undermine plasticity in continual learning. By introducing a loss-sensitive regularizer alongside the cross-entropy loss, we steer update directions so that post-projection distortion is minimized. Our theoretical analysis demonstrated that the loss-sensitive scaling parameter can better characterize loss perturbations on previous tasks. Empirical results on image-classification benchmarks showed that our two-stage gradient-projection method outperformed other projection and regularization approaches in balancing plasticity and stability. In future work, we will (i) investigate memory-efficient curvature approximations, (ii) modify the optimization stage to reduce runtime.

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

## A    APPENDIX

**Notation**    Let $f_W$ denote the network with parameters $W = \{\theta\}_{\ell=1}^L$. We denote per-layer as $\ell$ and task loss $L_t^\ell(\theta)$ of $t$-th task. Gradients are $g_t = \nabla_\theta L_t(\theta)$ and Hessians $H_t = \nabla_\theta^2 L_t(\theta)$. For a subspace $\mathcal{M}$, $\text{proj}_{\mathcal{M}^\perp}(gt)$ is the orthogonal projection of $gt$ onto $\mathcal{M}$. After completing task $t$, the global feature space is denoted by $\bar{M}(t)$.

## B    PROOFS AND DERIVATIONS

**Lemma B.1.** *Fix a task $t$ and a layer $\ell$. Let $\bar{M}^\ell(t-1)$ be the protected space accumulated up to task $t-1$, and let $S^\ell$ be the (layerwise) scaling operator acting on $\bar{M}^\ell(t-1)$. Denote by $g_t^\ell \in \mathbb{R}^{d_\ell}$ the cross-entropy gradient at layer $\ell$, and by $\mathcal{M}_j^\ell \subseteq \bar{M}^\ell(t-1)$ the feature space associated with a previous task $j$. Define the $S$-weighted update*

$$\delta\theta^\ell \;:=\; Proj_{\bar{M}^\ell(t-1)^\perp}^{S^\ell}\big(g_t^\ell\big).$$

*Assume that the leakage component $\big(Proj_{\bar{M}^\ell(t-1)^\perp}^{S^\ell} - Proj_{\bar{M}^\ell(t-1)^\perp}\big)(g_t^\ell)$, which lies in $\bar{M}^\ell(t-1)$, is colinear with $Proj_{\mathcal{M}_j^\ell}(g_t^\ell)$ (e.g., $S^\ell$ is isotropic on $\mathcal{M}_j^\ell$). Then there exists a scalar $p_\ell \in \mathbb{R}$ such that*

$$Proj_{\mathcal{M}_j^\ell}\big(\delta\theta^\ell\big) \;=\; p_\ell \, Proj_{\mathcal{M}_j^\ell}(g_t^\ell), \tag{11}$$

*where $p_\ell$ can be written as*

$$p_\ell \;:=\; \frac{\big\langle Proj_{\mathcal{M}_j^\ell}(\delta\theta^\ell),\, Proj_{\mathcal{M}_j^\ell}(g_t^\ell) \big\rangle}{\big\| Proj_{\mathcal{M}_j^\ell}(g_t^\ell) \big\|_2^2}, \tag{12}$$

*i.e., $p_\ell$ measures the fraction of $g_t^\ell$ that remains in the unscaled space $\bar{M}^\ell(t-1)$ (and hence can re-enter $\mathcal{M}_j^\ell$) after projecting $g_t^\ell$ onto the orthogonal complement of the scaled space $S^\ell \bar{M}^\ell(t-1)$.*

*Proof.* By definition of the $S$-weighted projection,

$$\delta\theta^\ell \;=\; Proj_{\bar{M}^\ell(t-1)^\perp}^{S^\ell}(g_t^\ell) \;=\; \underbrace{Proj_{\bar{M}^\ell(t-1)^\perp}(g_t^\ell)}_{\in \bar{M}^\ell(t-1)^\perp} + \underbrace{\Big(Proj_{\bar{M}^\ell(t-1)^\perp}^{S^\ell} - Proj_{\bar{M}^\ell(t-1)^\perp}\Big)(g_t^\ell)}_{\in \bar{M}^\ell(t-1)}.$$

Projecting both sides onto $\mathcal{M}_j^\ell \subseteq \bar{M}^\ell(t-1)$ yields

$$Proj_{\mathcal{M}_j^\ell}(\delta\theta^\ell) \;=\; \underbrace{Proj_{\mathcal{M}_j^\ell}\Big(Proj_{\bar{M}^\ell(t-1)^\perp}(g_t^\ell)\Big)}_{=\,0} + Proj_{\mathcal{M}_j^\ell}\Big(\big(Proj_{\bar{M}^\ell(t-1)^\perp}^{S^\ell} - Proj_{\bar{M}^\ell(t-1)^\perp}\big)(g_t^\ell)\Big).$$

By the colinearity assumption, the rightmost term is a scalar multiple of $Proj_{\mathcal{M}_j^\ell}(g_t^\ell)$, i.e.,

$$Proj_{\mathcal{M}_j^\ell}(\delta\theta^\ell) \;=\; p_\ell \, Proj_{\mathcal{M}_j^\ell}(g_t^\ell),$$

for some $p_\ell \in \mathbb{R}$. Taking the inner product with $Proj_{\mathcal{M}_j^\ell}(g_t^\ell)$ and normalising by its squared norm gives the explicit expression equation 13 for $p_\ell$, which proves Eq equation 11:

$$p_\ell \;=\; \frac{\big\langle Proj_{\mathcal{M}_j^\ell}\big(\big(Proj_{\bar{M}^\ell(t-1)^\perp}^{S^\ell} - Proj_{\bar{M}^\ell(t-1)^\perp}\big)(g_t^\ell)\big) \, Proj_{\mathcal{M}_j^\ell}(g_t^\ell) \big\rangle}{\big\| Proj_{\mathcal{M}_j^\ell}(g_t^\ell) \big\|_2^2} \tag{13}$$

$$=\; \frac{\big\langle Proj_{\mathcal{M}_j^\ell}(\delta\theta^\ell),\, Proj_{\mathcal{M}_j^\ell}(g_t^\ell) \big\rangle}{\big\| Proj_{\mathcal{M}_j^\ell}(g_t^\ell) \big\|_2^2}. \tag{14}$$

$\square$

**Theorem B.2** (Theorem 1 in Sec. 3.2 of the main text). *During the training of task $t$, fix any layer $\ell$. Let $\bar{M}^\ell(t-1)$ be the protected (total) feature space accumulated up to task $t-1$, let $S^\ell$ be the layerwise scaling operator, and let $g_t^\ell$ be the cross-entropy gradient at layer $\ell$. For any previous task $j$ with loss $L_j(\theta)$ and feature subspace $\mathcal{M}_j^\ell \subseteq \bar{M}^\ell(t-1)$, consider the actual update*

$$\delta\theta^\ell \;=\; Proj_{\bar{M}^\ell(t-1)^\perp}^{S^\ell}(g_t^\ell).$$

*Assume that $Proj_{\mathcal{M}_j^\ell}(\delta\theta^\ell)$ is colinear with $Proj_{\mathcal{M}_j^\ell}(g_t^\ell)$ (e.g., $S^\ell$ is isotropic on $\mathcal{M}_j^\ell$), and define the scalar*

$$p_\ell \;:=\; \frac{\left\langle Proj_{\mathcal{M}_j^\ell}(\delta\theta^\ell),\, Proj_{\mathcal{M}_j^\ell}(g_t^\ell) \right\rangle}{\left\| Proj_{\mathcal{M}_j^\ell}(g_t^\ell) \right\|_2^2}.$$

*Approximating the Hessian of $L_j$ at layer $\ell$ by the Fisher information $F_j$, the change in $L_j$ caused by applying $\{\delta\theta^\ell\}_\ell$ while training task $t$ is*

$$\Delta L_j \;=\; \sum_\ell p_\ell^2 \left( Proj_{\mathcal{M}_j^\ell}(g_t^\ell) \right)^\top F_j \, Proj_{\mathcal{M}_j^\ell}(g_t^\ell). \tag{15}$$

*Equivalently, $p_\ell$ measures the component of $g_t^\ell$ that remains in the unscaled space $\bar{M}^\ell(t-1)$ (and hence can re-enter $\mathcal{M}_j^\ell$) after projecting $g_t^\ell$ onto the orthogonal complement of the scaled space $S^\ell \bar{M}^\ell(t-1)$.*

*Proof.* Let $\theta_j$ denote the parameters after finishing task $j$. Consider the small update $\delta\theta = \{\delta\theta^\ell\}_\ell$ applied during task $t$. A second-order Taylor expansion of $L_j$ around $\theta_j$ gives

$$L_j(\theta_j + \delta\theta) - L_j(\theta_j) = \underbrace{\langle \nabla L_j(\theta_j),\, \delta\theta \rangle}_{=0} + \tfrac{1}{2} \sum_\ell \left( \delta\theta^\ell \right)^\top H_j^\ell \, \delta\theta^\ell + o(\|\delta\theta\|^2),$$

where $\nabla L_j(\theta_j) = 0$. For cross-entropy losses, it is standard to approximate $H_j^\ell$ by the Fisher information $F_j$ (up to a constant factor that can be absorbed). Moreover, in many continual-learning constructions the curvature of $L_j$ concentrates on the protected subspace $\mathcal{M}_j^\ell$ and the cross-terms with $(\mathcal{M}_j^\ell)^\perp$ are negligible, so that

$$\left( \delta\theta^\ell \right)^\top F_j \, \delta\theta^\ell \;\approx\; \left( Proj_{\mathcal{M}_j^\ell}(\delta\theta^\ell) \right)^\top F_j \, Proj_{\mathcal{M}_j^\ell}(\delta\theta^\ell).$$

Summing over $\ell$ yields

$$\Delta L_j \;\approx\; \tfrac{1}{2} \sum_\ell \left( Proj_{\mathcal{M}_j^\ell}(\delta\theta^\ell) \right)^\top F_j \, Proj_{\mathcal{M}_j^\ell}(\delta\theta^\ell).$$

By the colinearity assumption there exists a scalar $p_\ell$ such that

$$Proj_{\mathcal{M}_j^\ell}(\delta\theta^\ell) \;=\; p_\ell \, Proj_{\mathcal{M}_j^\ell}(g_t^\ell),$$

with $p_\ell$ given explicitly in the theorem statement. Substituting this relation into the quadratic form and absorbing the factor $\frac{1}{2}$ into $F_j$ (or redefining $\Delta L_j$ accordingly) we obtain

$$\Delta L_j = \sum_\ell p_\ell^2 \left( Proj_{\mathcal{M}_j^\ell}(g_t^\ell) \right)^\top F_j \, Proj_{\mathcal{M}_j^\ell}(g_t^\ell),$$

which is exactly Eq equation 15. $\qquad\square$

## C ALGORITHMIC DETAILS

In this section, we present the detailed construction of the feature spaces used in Step 1 of our two-stage gradient-projection method, providing an expanded account of Step 1 in the main text (Sec. 3.1), including how to select the top $r$ orthonormal basis vectors $\{u_{i_1}, \ldots, u_{i_r}\}$ via a threshold $\epsilon$, how to construct the task-specific subspace $\mathcal{M}_{t-1}$, and how to update the global feature space $\bar{M}(t-2)$ once training on task $t-1$ is complete. We then present the pipeline illustrating the construction of the loss-sensitive space. Finally, we provide the full algorithmic procedure as Algorithm 2.

### C.1 A SVD AND $k$-RANK APPROXIMATION

Singular Value Decomposition (SVD) can be used to factorize a rectangular matrix $R = U\Sigma V^\top \in \mathbb{R}^{m \times n}$ into the product of three matrices, where $U \in \mathbb{R}^{m \times m}$ and $V \in \mathbb{R}^{n \times n}$ are orthonormal, and $\Sigma$ is a diagonal matrix containing the singular values sorted along its main diagonal. If the rank of $R$ is $r \leq \min(m, n)$, then $R = \sum_{i=1}^{r} \sigma_i u_i v_i^\top$, where $u_i$ and $v_i$ are the left and right singular vectors and $\sigma_i \in \mathrm{diag}(\Sigma)$ are the singular values. A $k$-rank approximation of $R$ can be written as $R_k = \sum_{i=1}^{k} \sigma_i u_i v_i^\top$ with $k \leq r$, where $k$ is chosen as the smallest index satisfying $\|R_k\|_F^2 \geq \epsilon_{\mathrm{th}} \|R\|_F^2$. Here, $\|\cdot\|_F$ denotes the Frobenius norm and $\epsilon_{\mathrm{th}} \in (0, 1)$ is the threshold hyperparameter.

### C.2 CONSTRUCTING THE TASK–SPECIFIC SUBSPACE $\mathcal{M}_{t-1}^\ell$

Let the global feature space accumulated up to task $t-2$ be $\bar{M}^\ell(t-2) = [u_1^\ell, \ldots, u_r^\ell] \in \mathbb{R}^{d \times r}$ with orthonormal columns, and let $R_{t-1}^\ell \in \mathbb{R}^{d \times N}$ denote the representation matrix extracted from data of task $t-1$ at layer $\ell$. We select the most informative directions for task $t-1$ by combining (i) the portion of $R_{t-1}^\ell$ that lies in the old global space $\bar{M}^\ell(t-2)$ and (ii) the portion that is orthogonal to it.

*(i) Energy inside the old global space* For each basis $u_i^\ell$ of $\bar{M}^\ell(t-2)$, compute its contribution to $R_{t-1}^\ell$ as $\delta_i^\ell = \|(u_i^\ell)^\top R_{t-1}^\ell\|_2^2 = (u_i^\ell)^\top R_{t-1}^\ell (R_{t-1}^\ell)^\top u_i^\ell$. Large $\delta_i^\ell$ indicates that the corresponding old direction is important for the current task.

*(ii) Energy beyond the old global space* Remove the component of $R_{t-1}^\ell$ already captured by $\bar{M}^\ell(t-2)$ via $\hat{R}_{t-1}^\ell = R_{t-1}^\ell - \bar{M}^\ell(t-2)\bar{M}^\ell(t-2)^\top R_{t-1}^\ell$, and compute its thin SVD $\hat{R}_{t-1}^\ell = \hat{U}^\ell \hat{\Sigma}^\ell (\hat{V}^\ell)^\top$. The squared singular values $\hat{\sigma}_h^{\ell\,2}$ quantify the energy of novel directions $\hat{u}_h^\ell$ that are orthogonal to $\bar{M}^\ell(t-2)$.

*(iii) Joint selection* Form a single score vector by concatenation $\delta = (\delta_1^\ell, \ldots, \delta_r^\ell, \hat{\sigma}_1^{\ell\,2}, \ldots, \hat{\sigma}_m^{\ell\,2})$ and sort it in descending order to get $\delta_{(1)} \geq \delta_{(2)} \geq \cdots$. Choose the smallest $k_{t-1}^\ell$ such that $\sum_{i=1}^{k_{t-1}^\ell} \delta_{(i)} \geq \epsilon_{\mathrm{th}} \|R_{t-1}^\ell\|_F^2$ with $\epsilon_{\mathrm{th}} \in (0, 1)$. Let $I_{\mathrm{old}}$ be the indices among the top-$k_{t-1}^\ell$ that come from $\{\delta_i^\ell\}$ and $I_{\mathrm{nov}}$ those that come from $\{\hat{\sigma}_h^{\ell\,2}\}$. The task–specific subspace is then $\mathcal{M}_{t-1}^\ell = [[u_i^\ell]_{i \in I_{\mathrm{old}}}, [\hat{u}_h^\ell]_{h \in I_{\mathrm{nov}}}]$, optionally followed by an orthonormalization step. This $\mathcal{M}_{t-1}^\ell$ captures both the reused directions from the previous global space and the novel directions required by task $t-1$.

### C.3 UPDATE FEATURE SPACE $\bar{M}(t-2)$

To obtain the updated global feature space $\bar{M}^\ell(t-1)$ after learning task $t-1$, we start from the previous global space $\bar{M}^\ell(t-2)$ and the task's representation $R_{t-1}^\ell \in \mathbb{R}^{d \times N}$. We extract the task-specific subspace $\mathcal{M}_{t-1}^\ell$ (see Constructing the task–specific subspace), then we update $\bar{M}(t-2)$, that is

$$\bar{M}^\ell(t-1) = \bar{M}^\ell(t-2) \oplus \mathcal{M}_{t-1}^\ell,$$

i.e., by taking the column span of $[\bar{M}^\ell(t-2), \mathcal{M}_{t-1}^\ell]$ and orthonormalizing once. *Equivalently*, since $\mathcal{M}_{t-1}^\ell$ contains reused and novel directions, in practice we only append the novel bases $\{\hat{u}_h^\ell\}_{h \in I_{\mathrm{nov}}}$: $\bar{M}^\ell(t-1) = \mathrm{span}([\bar{M}^\ell(t-2), \hat{u}_h^\ell]_{h \in I_{\mathrm{nov}}})$.

---

**Algorithm 2** Dual-Stage Gradient Projection Algorithm

---

**Require:** Task stream $\mathcal{T} = \{\mathcal{D}_1, \ldots, \mathcal{D}_T\}$; network $f_W$ with $L$ layers; learning rate $\eta$; energy threshold $\epsilon$; scale coefficient $\alpha$; soft weight $\lambda$;
**Ensure:** Trained weights $W_T = \{\theta^\ell\}_{\ell=1}^L$
1: $\bar{M}^\ell \leftarrow \varnothing$ {protected basis on each layer $\ell$}
2: $S_{lss}^\ell, S_{sgp}^\ell \leftarrow I$ {scaling matrix of each layer $\ell$}
3: $memory \leftarrow \varnothing$
4: **for** $t = 1$ **to** $T$ **do**
5:     // Begin Training Loop (Main text Sec.3.1 Step 2)
6:     **while** not converged on $\mathcal{D}_t$ **do**
7:         Sample minibatch $B_t \subset \mathcal{D}_t$
8:         $g_t \leftarrow \nabla_W L_{\text{CE}}(B_t; W)$
9:         $L_{\text{total}} \leftarrow L_{\text{CE}}(B_t; W) + \lambda \sum_\ell \|\text{Proj}_{\bar{M}^\ell}^{S_{lss}^\ell}(g_t)\|_2^2$       $\triangleright$ *Main text Sec.3.2 Eq. (4)*
10:        $g_S \leftarrow \nabla_W L_{\text{total}}$
11:        **for** $\ell = 1$ **to** $L$ **do**
12:            $g_{\text{SH},\ell} \leftarrow \text{Proj}_{\bar{M}^\ell,\perp}^{S_{sgp}^\ell}(g_{S,\ell})$          $\triangleright$ *Main text Sec.3.2 Eq. (9)*
13:            $\theta^\ell \leftarrow \theta^\ell - \eta\, g_{\text{SH},\ell}$                $\triangleright$ *Main text Sec.3.2 Eq. (10)*
14:        **end for**
15:     **end while**
16:     // Update Protected Feature Space (Main text Sec.3.1 Step 1)
17:     **for** $\ell = 1$ **to** $L$ **do**
18:         $g_f = \nabla_W L_{CE}(D_t; W)$
19:         Construct fisher matrix $F_t \leftarrow g_f^2$
20:         Sample $n_s$ activations $R_t^\ell$
21:         $\bar{\Sigma} = \|(M^\ell(t-1))^\top R_t^\ell\|_2^2$          $\triangleright$ *Supplment Sec.3.2 Step (i)*
22:         $\hat{R}_t^\ell = R_t^\ell - \bar{M}^\ell (\bar{M}^\ell)^\top R_t^\ell$        $\triangleright$ *Supplment Sec.3.2 Step (ii)*
23:         $(\hat{U}, \hat{\Sigma}) \leftarrow \text{SVD}(\hat{R}_t^\ell)$               $\triangleright$ *Supplment Sec.3.2 Step(ii)*
24:         $k \leftarrow$ smallest $k$ s.t. $\|[\hat{\Sigma}, \bar{\Sigma}]_{1:k}\|_F^2 \geq \epsilon\|[\hat{\Sigma}, \bar{\Sigma}]\|_F^2$    $\triangleright$ *Supplment Sec.3.2 Step(iii)*
25:         Get $\mathcal{M}_t^\ell$ by *Supplment Sec.3.2 Step (iii)* and $k$
26:         $\bar{M}^\ell \leftarrow \bar{M}^\ell \oplus \mathcal{M}_t^\ell$                  $\triangleright$ *Supplment Sec.3.3*
27:         $(U, \Sigma) \leftarrow \text{SVD}(R_t^\ell)$
28:         $S_{sgp}^\ell = SGP(S_{sgp}^\ell; \Sigma)$         $\triangleright$ *Constructing the SGP Scaling Matrix*
29:        **for all** new basis $u_{i,t} \in \bar{M}_\ell$ **do**
30:            $\text{LSW}(u_{i,t}^\ell) = \sum_{j=1}^t \text{Proj}_{\mathcal{M}_j^\ell}(u_{i,t}^\ell)^\top F_j \text{Proj}_{\mathcal{M}_j^\ell}(u_{i,t}^\ell).$   $\triangleright$ *Main text Sec.3.2 Eq. (6)*
31:            $s_{i,t} = \frac{(1+\alpha)\,\text{LSW}(u_{i,t}^\ell)}{\alpha\,\text{LSW}(u_{i,t}^\ell) + \max_m \text{LSW}(u_{m,t}^\ell)},$       $\triangleright$ *Main text Sec.3.2 Eq. (7)*
32:        **end for**
33:        $\bar{S}_{lss}^\ell \leftarrow \text{diag}(s_1, \ldots, s_{|\bar{M}^\ell|})$        $\triangleright$ *Main text Sec.3.2 Eq. (8)*
34:        $memory \leftarrow F_t, \mathcal{M}_t$
35:     **end for**
36: **end for**
37: **return** $W$

---

## C.4   Loss-Sensitive Space Pipeline

In this section, we illustrate the *loss-sensitive space* (LSS) construction pipeline used at the end of task $t-1$. Figure 5 depicts the entire procedure for transforming each basis vector $u_i$ of the $k$-dimensional orthonormal space $\bar{M}(t-1) = [u_1, \ldots, u_k]$ (drawn inside the unit circle) into its scaled counterpart $u_i^{\text{new}}$. The process consists of the following steps:

**1. Basis decomposition** Decompose the protected feature space into its individual directions $\{u_i\}_{i=1}^k$.

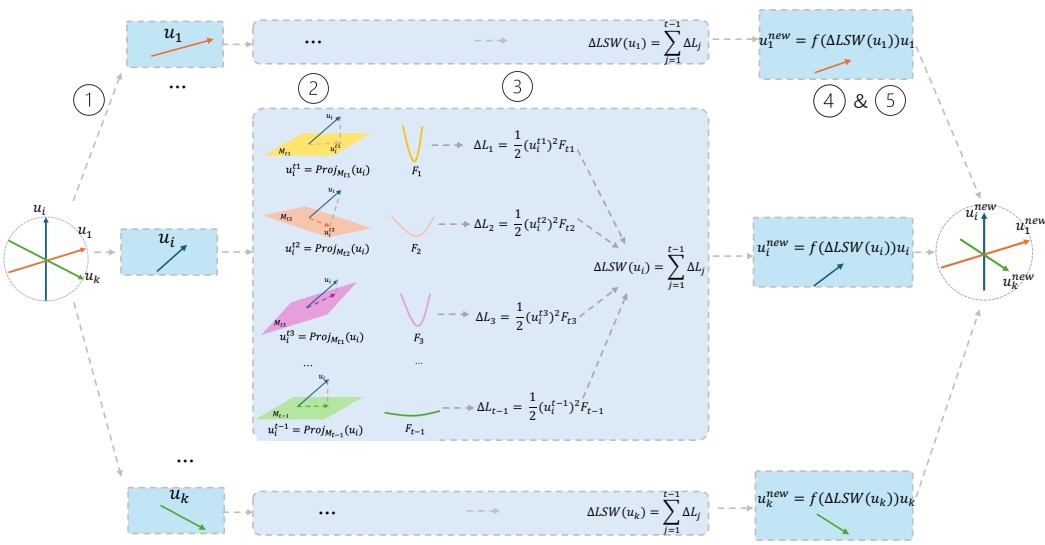

Figure 5: Loss-Sensitive Space Pipeline: (1) The arrows inside the unit circle denote different basis vectors of the feature subspace $\bar{M}(t-1)$; (2) the colored parallelograms represent the past tasks' task-specific feature spaces $M_{1:t-1}$, and the colored curves indicate task curvature information, i.e., the Fisher matrices $F_j$.

**2. Task-wise projection** For each direction $u_i$, we project it onto every previous task-specific subspace $M_j$ ($j = 1, \ldots, t-1$) to determine how a unit-length perturbation $\Delta\theta$ along $u_i$ affects the stored parameters $\theta_j$.

**3. Loss sensitivity per task** Incorporate the curvature of each past task via its Fisher matrix $F_j$ and compute

$$\Delta L_j(u_i) \; = \; (Proj_{M_j}(u_i))^\top F_j Proj_{M_j}(u_i),$$

which estimates the loss increase of task $j$ caused by a unit-length move along $u_i$. Averaging over all past tasks yields the *loss-sensitivity weight*

$$\Delta LSW(u_i) \; = \; \frac{1}{t-1} \sum_{j=1}^{t-1} \Delta L_j(u_i).$$

**4. Normalization and scaling matrix** Apply a normalization function $f$ that maps the values $\Delta LSW(u_i)$ to the interval $[0, 1]$, producing the scaling coefficients $s_i = f(\Delta LSW(u_i))$, where $f$ is Eq. (7) in the main text. Collect them in the diagonal matrix $S_{\mathrm{lss}} = \mathrm{diag}(s_1, \ldots, s_k)$.

**5. Constructing the feature space LSS** Finally, scale the original orthonormal basis to obtain the loss-sensitive space $\bar{M}_{\mathrm{LSS}}(t-1) = [s_1 u_1, \; s_2 u_2, \; \ldots, \; s_k u_k]$, which is used in Stage 1 of our two-stage gradient-projection algorithm.

## D    EXPERIMENTAL PROTOCOL

In this section, we give the statistics of three datasets applied to conduct experiments in Table 3. In addition, the settings of hyperparameters for all the considered methods are demonstrated in Table 4. Where CIFAR-100, Superclass and MiniImageNet denote 10-Split CIFAR-100, 20-Split CIFAR-100 Superclass and 20-Split MiniImageNet respectively. Finally, we provide supplementary results for the "plasticity and stability analysis" and "parameter sensitivity analysis" experiments presented in the main text.

### D.1 DATASETS AND SPLITS

We list datasets (e.g., CIFAR-100 10-split, MiniImageNet), class orders, samples per task, and any randomization rules.

Table 3: Statistics of the three benchmarks used in our experiments.

|  | 10-Split CIFAR-100 | 20-Split CIFAR-100 | 20-Split MiniImageNet |
|---|---|---|---|
| Total Number of Tasks | 10 | 20 | 20 |
| Total Number of Classes | 100 | 100 | 100 |
| Size of Input Data | $3 \times 32 \times 32$ | $3 \times 32 \times 32$ | $3 \times 84 \times 84$ |
| Number of Classes / Task | 10 | 5 | 5 |
| Sample Size of Training Set / Task | 4750 | 2375 | 2450 |
| Sample Size of Valid Set / Task | 250 | 125 | 50 |
| Sample Size of Test Set / Task | 1000 | 500 | 500 |

Unless otherwise stated, we use the repository's default backbone (kept fixed across tasks) and only expand the final linear classifier as classes accumulate. All runs use SGD with momentum 0.9 and the same data preprocessing as in the main paper. Steps per epoch are computed as $\lceil N_{\text{train}}/B \rceil$ with batch size $B$; iterations per task are (steps/epoch) $\times$ (epochs).

**10-Split CIFAR-100**   Optimizer: *SGD (momentum 0.9)*. Initial learning rate: *0.05*. Scheduler: *Reduce-on-Plateau* on the validation metric with *patience* = 7, *factor* = 2 (i.e., LR is divided by 2 when the metric plateaus), and minimum LR $10^{-4}$. Batch sizes: *64/64* for train/test. Each task is trained for *200 epochs*. With $N_{\text{train}} = 4750$ and $B = 64$, steps/epoch = $\lceil 4750/64 \rceil = 75$, yielding *about 15,000 iterations per task*.

**20-Split CIFAR-100 Superclass**   Optimizer: *SGD (momentum 0.9)*. Initial learning rate: *0.01*. Scheduler: *Reduce-on-Plateau* with *patience* = 6, *factor* = 2, and minimum LR $10^{-5}$. Batch sizes: *64/64*. Each task is trained for *50 epochs*. With $N_{\text{train}} = 2375$ and $B = 64$, steps/epoch = $\lceil 2375/64 \rceil = 38$, giving *about 1,900 iterations per task*.

**20-Split MiniImageNet**   Optimizer: *SGD (momentum 0.9)*. Initial learning rate: *0.1*. Scheduler: *Reduce-on-Plateau* with *patience* = 5, *factor* = 3, and minimum LR $10^{-3}$. Batch sizes: *64/64*. Each task is trained for *100 epochs*. With $N_{\text{train}} = 2450$ and $B = 64$, steps/epoch = $\lceil 2450/64 \rceil = 39$, resulting in *about 3,900 iterations per task*.

### D.2 HYPERPARAMETERS

The settings of hyperparameters for all the considered methods are demonstrated in Tabel 4. Since both LSS and SGP require a hyperparameter to adjust the scaling in Eq. equation 16, we denote this hyperparameter by $\alpha_{\text{SGP}}$ when the method is SGP and by $\alpha_{\text{LSS}}$ when the method is LSS.

### D.3 PLASTICITY AND STABILITY

In this section we analyse the plasticity and stability of our method on all benchmarks by comparing, for each task, the *first-pass accuracy* (1st-ACC) and the *backward transfer* (BWT) under different training objectives. We consider the following two settings:

1. **Cross-Entropy Only** The model is trained using only the cross-entropy loss, which maximises plasticity but provides the lowest stability:

$$g_t^\ell = \nabla_{w^\ell} \mathcal{L}_{\text{CE}}(W, \mathbb{D}_t), \qquad \mathcal{L} = \mathcal{L}_{\text{CE}}.$$

2. **Cross-Entropy + LSS soft regulariser** The training objective augments the cross-entropy with an LSS-based soft regulariser that penalises the gradient component inside the protected subspace:

$$\mathcal{L} = \mathcal{L}_{\text{CE}} + \lambda \sum_\ell \left\| \text{Proj}_{M^\ell}^{S_{\text{lss}}}(g_t^\ell) \right\|_2^2,$$

where the soft-regulariser hyperparameters are chosen to achieve the best results (see Table-3 for details).

Table 4: List of hyperparameter settings in baseline approaches and our methods. Here, *lr* denotes the initial learning rate, and $n_s$ is the number of samples drawn from previous tasks to construct the projection space for the current task.

| Methods | Hyperparameter Settings |
|---|---|
| Multitask | *lr*: 0.05 (CIFAR-100), 0.01 (Superclass), 0.1 (MiniImageNet). |
| OWM | *lr*: 0.01 (CIFAR-100), 0.1 (MiniImageNet). |
| A-GEM | *lr*: 0.05 (CIFAR-100, Superclass), 0.1 (MiniImageNet); memory size (samples): 2000 (CIFAR-100, Superclass), 500 (MiniImageNet). |
| ER_Res | *lr*: 0.05 (CIFAR-100, Superclass), 0.1 (MiniImageNet). |
| Adam-NSCL | *lr*: $10^{-4}$ (CIFAR-100, Superclass), $5 \times 10^{-5}$ (MiniImageNet). |
| GPM | *lr*: 0.01 (CIFAR-100, Superclass), 0.1 (MiniImageNet); $n_s$: 125 (CIFAR-100, Superclass), 100 (MiniImageNet). |
| FS-DGPM | *lr*, $\eta_3$: 0.01 (CIFAR-100, Superclass), 0.1 (MiniImageNet); *lr* for sharpness, $\eta_1$: 0.001 (CIFAR-100), 0.01 (Superclass, MiniImageNet); *lr* for DGPM, $\eta_2$: 0.01 (CIFAR-100, Superclass, MiniImageNet); memory size (samples): 1000 (CIFAR-100, Superclass, MiniImageNet); $n_s$: 125 (CIFAR-100, Superclass), 100 (MiniImageNet). |
| CGP | *lr*: 0.04 (CIFAR-100), 0.03 (Superclass), 0.1 (MiniImageNet); $n_s$: 125 (CIFAR-100, Superclass), 100 (MiniImageNet). |
| TRGP | *lr*: 0.01 (CIFAR-100, Superclass), 0.1 (MiniImageNet); $n_s$: 125 (CIFAR-100, Superclass), 100 (MiniImageNet). |
| SGP | *lr*: 0.05 (CIFAR-100), 0.01 (Superclass), 0.1 (MiniImageNet); $n_s$: 125 (CIFAR-100, Superclass), 100 (MiniImageNet); $\alpha$: 5 (CIFAR-100), 3 (Superclass), 1 (MiniImageNet). |
| GPCNS | *lr*: 0.05 (CIFAR-100), 0.01 (Superclass), 0.1 (MiniImageNet); $\alpha$: 5 (CIFAR-100), 4.5 (Superclass), 3 (MiniImageNet). |
| GPM + GPCNS | *lr*: 0.05 (CIFAR-100), 0.01 (Superclass), 0.1 (MiniImageNet); $\alpha$: 1.5 (CIFAR-100), 4.5 (Superclass), 1 (MiniImageNet); $n_s$: 125 (CIFAR-100, Superclass), 100 (MiniImageNet). |
| TRGP + GPCNS | *lr*: 0.05 (CIFAR-100), 0.01 (Superclass), 0.1 (MiniImageNet); $\alpha$: 1.5 (CIFAR-100), 4.5 (Superclass), 1 (MiniImageNet); $n_s$: 125 (CIFAR-100, Superclass), 100 (MiniImageNet). |
| SGP + GPCNS | *lr*: 0.05 (CIFAR-100), 0.01 (Superclass), 0.1 (MiniImageNet); $\alpha$: 1.5 (CIFAR-100), 4.5 (Superclass), 1 (MiniImageNet); $n_s$: 125 (CIFAR-100, Superclass), 100 (MiniImageNet). |
| LSS + SGP | *lr*: 0.05 (CIFAR-100), 0.01 (Superclass), 0.1 (MiniImageNet); $\alpha_{LSS}$: 10 (CIFAR-100), 10 (Superclass), 5 (MiniImageNet); $\alpha_{SGP}$: 10 (CIFAR-100), 3 (Superclass), 5 (MiniImageNet); $\lambda$: 1e0 (CIFAR-100, Superclass, MiniImageNet) $n_s$: 125 (CIFAR-100, Superclass), 100 (MiniImageNet). |
| LSS + TRGP | *lr*: 0.05 (CIFAR-100), 0.01 (Superclass), 0.1 (MiniImageNet); $\lambda$: 1e0 (CIFAR-100, Superclass, MiniImageNet) $n_s$: 125 (CIFAR-100, Superclass), 100 (MiniImageNet). |

We further provide a sensitivity analysis for the loss-sensitive scaling (LSS) parameters. With hyperparameter $\alpha$, each basis scaling factor (cf. Eq. (7) in the main text) is

$$s_{i,t-1} = \frac{(1+\alpha)\,\mathrm{LSW}\big(\bar{u}_{i,t-1}^{\ell}\big)}{\alpha\,\mathrm{LSW}\big(\bar{u}_{i,t-1}^{\ell}\big) + \max_m \mathrm{LSW}\big(\bar{u}_{m,t-1}^{\ell}\big)}, \tag{16}$$

and the scaling matrix is

$$S_{\mathrm{lss}}^{\ell} = \mathrm{diag}\big(s_{1,t-1}, \ldots, s_{k,t-1}\big). \tag{17}$$

From the left subpanel of Figure 6(a)–(b) and (c), we observe that adding the soft regulariser does not materially harm plasticity: the 1st-ACC remains essentially on par with the Cross-Entropy-only setting. In contrast, from the right subpanels of Figure 6(a)–(c), we see that as $\alpha$ increases, accuracy gradually drops. Under the same $\lambda$, the *unit-orthonormal* variant (denoted by "$+\infty$", i.e. $S_{\mathrm{lss}} = I$) imposes a stronger constraint and causes a more severe performance drop.

These results indicate that, with a moderate $\alpha$, the soft regulariser preserves plasticity while reducing the distortion introduced by the subsequent hard projection. However, as $\alpha \to \infty$, the scaling fac-

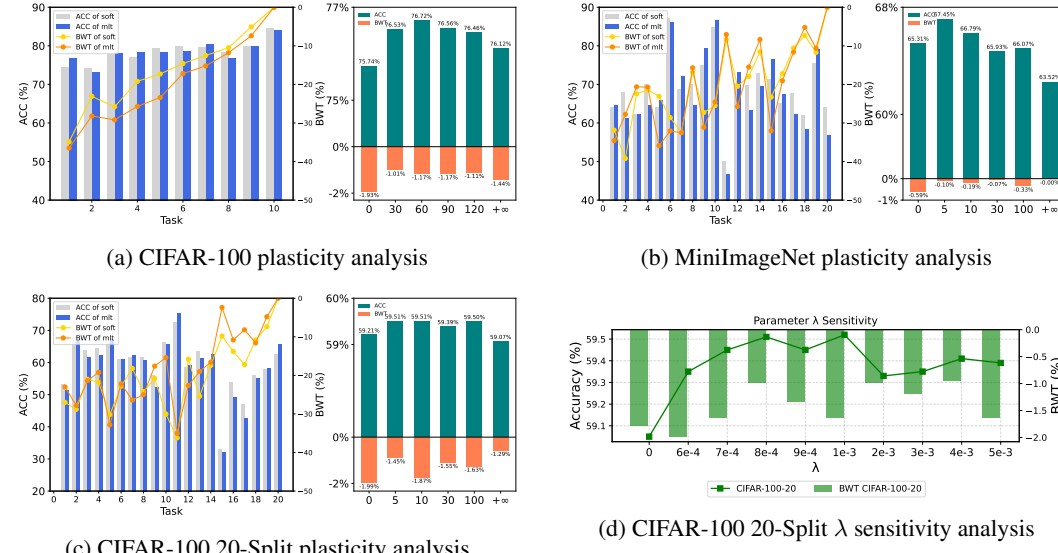

(a) CIFAR-100 plasticity analysis

(b) MiniImageNet plasticity analysis

(c) CIFAR-100 20-Split plasticity analysis

(d) CIFAR-100 20-Split $\lambda$ sensitivity analysis

Figure 6: Four subplots showing (a) CIFAR-100 plasticity analysis, (b) MiniImageNet plasticity analysis, (c) CIFAR-100 20-Split plasticity analysis, and (d) CIFAR-100 20-Split $\lambda$ sensitivity analysis.

tors approach 1, the constraint on $g_t$ becomes overly strong, and performance degrades—plasticity suffers most in the unit-orthonormal limit. Thus, a suitable $\alpha$ achieves minimal projection-induced distortion with almost no damage to plasticity.

### D.4 PARAMETER SENSITIVE ANALYSIS

In this section, as a supplement to the main paper's experiments, we analyze the sensitivity of the regularization weight $\lambda$ on the CIFAR-100 20-split benchmark. Our total training loss is defined as

$$\mathcal{L}_{\text{total}} = \mathcal{L}_{\text{CE}} + \lambda \, \mathcal{R}_{\text{soft}},$$

where $\mathcal{R}_{\text{soft}} = \left\| \text{Proj}_{\text{LSS}}(g_t) \right\|_2$. As shown in Figure 6(d), the best performance is achieved at $\lambda = 10^{-3}$, and overall the model remains stable for $\lambda$ in the range $[5 \times 10^{-3}, 6 \times 10^{-1}]$. This demonstrates that our method exhibits low parameter sensitivity on this dataset.

## E REPRODUCIBILITY CHECKLIST

### E.1 ENVIRONMENT

All experiments were conducted on a single Ubuntu 22.04 machine with 13th Gen Intel(R) Core(TM) i5-13600KF CPU and one NVIDIA GeForce RTX 4090 (24 GB; CUDA 11.8). Our code is implemented in Python 3.8.20 using PyTorch 2.2.0+cu118 and TorchVision 0.17.0+cu118. Unless otherwise specified, CUDA and cudnn versions are those bundled with the installed PyTorch build (reported as `torch.version.cuda` and `torch.backends.cudnn.version()`).

### E.2 CODE

The source code required to reproduce our experiments is bundled as the folder `code/` in the supplementary material. Please place the entire `code/` folder in the same directory and and keep its internal directory structure unchanged. To run:

1. Create a Python environment following `code/requirements.txt`.

2. From the root of `code/`, execute the main entry script, e.g.,

```
cd code
python LSS_cifar100.py
```

