# OpenReview forum: "Dual-Stage Gradient Projection Based Continual Learning: Enhancing Plasticity and Preserving Stability"
_ICLR.cc/2026/Conference — ICLR 2026 Conference Withdrawn Submission_

### Official Review · Reviewer_3g4e · 2025-10-29

**Soundness:** 3
**Presentation:** 3
**Contribution:** 2
**Rating:** 4
**Confidence:** 4

**Summary:**

This paper develops a new orthogonal gradient projection–based method, which adopts a two-stage approach: first, it uses a regularization constraint to encourage the gradient direction of the new task to align with the orthogonal complement space; second, it performs orthogonal projection to prevent task conflicts.

**Strengths:**

- This paper argue that existing orthogonal projection methods may harm the plasticity of new tasks when gradients are constrained to be strictly orthogonal.
- It proposes to design the gradient regularization term based on the Fisher Information Matrix.
- The paper is well-structured and clearly written.

**Weaknesses:**

- In each optimization step, this paper requires two rounds of gradient computation (two-stage process), which significantly increases time and memory costs. However, the paper lacks a detailed comparison of time and memory efficiency with existing orthogonal projection methods.
- All experiments are conducted on small-scale datasets such as CIFAR-100 and MiniImageNet, lacking validation on larger and more challenging datasets like ImageNet.
- The method is only verified on small or outdated architectures such as AlexNet, LeNet, and ResNet, without evaluation on Transformer-based architectures.
- The paper lacks sufficient discussion and comparison with related works [1–3].
- The font in Figure 4 is inconsistent with the main text.

[1]Liang, Yan-Shuo, and Wu-Jun Li. "Adaptive plasticity improvement for continual learning." Proceedings of the IEEE/CVF conference on computer vision and pattern recognition. 2023.

[2]Yang, E., Shen, L., Wang, Z., Liu, S., Guo, G., & Wang, X. (2023). Data augmented flatness-aware gradient projection for continual learning. In Proceedings of the IEEE/CVF international conference on computer vision (pp. 5630-5639).

[3]Yang, Z., Yang, Z., Liu, Y., Li, P., & Liu, Y. (2023). Restricted orthogonal gradient projection for continual learning. AI Open, 4, 98-110.

**Questions:**

- Does the regularization term in Equation (4) constrain the performance of the new task compared with directly using the cross-entropy loss?
- In Figure 4, why is the lambda hyperparameter set to such a large value? How is the balance between the regularization loss and the original cross-entropy loss achieved?

---

### Official Review · Reviewer_NrB1 · 2025-10-31

**Soundness:** 2
**Presentation:** 2
**Contribution:** 2
**Rating:** 4
**Confidence:** 3

**Summary:**

The paper presents a gradient-projection-based task-incremental learning method. The authors argue that the strict orthogonal projection used in existing methods causes deviations in parameter updates, which negatively affect plasticity. To address this issue, they propose a loss-sensitive space (LSS) regularization term that encourages gradients to update along directions approximately orthogonal to the feature space of previous tasks. Since LSS gradually aligns the parameters during training, the gradients become partially orthogonal in advance, making subsequent orthogonal gradient projection easier and less distortive. The proposed method is evaluated on CIFAR-100 and MiniImageNet using AlexNet, LeNet, and a reduced ResNet-18.

**Strengths:**

- The proposed method is well designed and supported by mathematical justification.

- The motivation is reasonable: direct gradient projection can indeed distort the optimal update direction.

**Weaknesses:**

- Although LSS regularization encourages orthogonal alignment and alleviates the drawbacks of hard gradient projection, it does not fully resolve the authors’ main concern about gradient deviation, since gradients must still be moved into the orthogonal space.

- The technical novelty is limited. The contribution lies primarily in introducing a single regularization term, which—while conceptually fine—is not particularly innovative or significant.

- The experimental design requires major improvement. The authors use AlexNet and LeNet, which, while historically important, are outdated. They should employ at least ResNet-18 for all datasets and compare results against baselines using that architecture. Ideally, more modern backbones such as transformers should also be tested.

- The performance results are not impressive. For example, [1] reports 95.3% accuracy on Split-CIFAR-10 with ResNet-18 for task-incremental learning, whereas the proposed method achieves only 78.05% using AlexNet. The authors should evaluate their approach on Split-CIFAR-100 using ResNet-18 and compare it with this state-of-the-art method to justify their contribution.

[1] A Theoretical Study on Solving Continual Learning

**Questions:**

Please refer to my comments in weaknesses

Additionally,

I think the gradient-projection method is inherently limited by the dimensionality of each subspace, which increases with the number of tasks. How efficiently does it utilize the subspace? Could you try evaluating it on a more challenging dataset such as Tiny-ImageNet, so that the proposed method is demonstrated on a problem with more classes per task and a larger number of tasks?

---

### Official Review · Reviewer_Gkbc · 2025-11-01

**Soundness:** 2
**Presentation:** 2
**Contribution:** 2
**Rating:** 2
**Confidence:** 4

**Summary:**

The paper proposes Dual-Stage Gradient Projection, a continual learning method aiming to improve plasticity in gradient-projection-based approaches while maintaining stability. The authors argue that conventional projection methods such as GPM and SGP cause strong gradient distortion when enforcing strict orthogonality to past-task feature subspaces. To address this, they introduce a two-stage scheme: a soft regularization stage that guides gradients toward directions approximately orthogonal to past subspaces using a loss-sensitive space (LSS), followed by a hard projection stage that applies a conventional orthogonal projection with minimal distortion. The LSS is computed using curvature information (approximated via Fisher matrices) from previous tasks to estimate the importance of each basis direction. Experiments on Split CIFAR-100, CIFAR-100 Superclass, and Split MiniImageNet show that proposed method improves average accuracy and reduces forgetting compared to several baselines such as SGP, TRGP, and GPM.

**Strengths:**

The paper tackles an important and well-known problem in continual learning: the plasticity–stability trade-off in gradient-projection methods. The general intuition of minimizing projection-induced distortion before enforcing orthogonality is conceptually appealing and addresses a legitimate limitation of prior work. The method is well-positioned within the literature, and the authors include a range of comparisons against relevant baselines, showing consistent accuracy improvements on multiple datasets. The idea of using a loss-sensitive scaling space to modulate gradient alignment through curvature information is theoretically interesting, and the overall framework appears compatible with existing projection-based algorithms.

**Weaknesses:**

The presentation of the paper is often unclear, and several crucial aspects of the method are underspecified or insufficiently justified. The role and necessity of the loss-sensitive space (LSS) are not well explained beyond formal equations. Theoretical analysis is largely superficial: the paper presents a theorem relating curvature information to projection scaling but provides almost no discussion or intuition about how this contributes to reducing gradient distortion or improving learning dynamics. Merely stating the theorem without elaborating on its implications or limitations leaves the reader uncertain about why the approach should work in practice.

The implementation details are also ambiguous. Since the proposed total loss includes a regularization term that depends on the gradient of the cross-entropy loss, it is unclear whether this formulation requires computing second-order derivatives (Hessian–vector products or Fisher estimates) during backpropagation. If so, this would entail a considerable computational and memory cost compared to first-order projection methods such as GPM or SGP. However, no analysis or empirical measurement of runtime or memory overhead is provided, which makes it difficult to assess the method’s practicality.

The ablation studies are confusing and lack sufficient explanation. For example, the “MTL + soft” configuration is described but not well motivated,  in multitask learning, all tasks are trained jointly, so it is unclear how a continual setting with “MTL + soft” is implemented or what it represents conceptually. Furthermore, while the paper reports numerical gains, it does not adequately analyze why the improvements occur or how sensitive they are to specific hyperparameters or architecture choices.
In addition, the paper omits important related work and citations. For instance, more recent methods that address gradient-space regularization or continual subspace updates (e.g., CODE-CL [1], DFGP[2], and others) are missing. This omission weakens the paper’s positioning in the current literature and makes it harder to assess its novelty relative to ongoing work.

Finally, the clarity and organization of the text need significant improvement. Many sections repeat material or present overly long mathematical derivations without intuitive commentary, while the motivation and implications of each design choice remain vague. The result is a paper that feels formally complex but conceptually underexplained.

[1] Apolinario, M.P., Choudhary, S. and Roy, K., 2025. CODE-CL: Conceptor-Based Gradient Projection for Deep Continual Learning. In Proceedings of the IEEE/CVF International Conference on Computer Vision (pp. 775-784).

[2] Yang, E., Shen, L., Wang, Z., Liu, S., Guo, G. and Wang, X., 2023. Data augmented flatness-aware gradient projection for continual learning. In Proceedings of the IEEE/CVF international conference on computer vision (pp. 5630-5639).

**Questions:**

•	How exactly is the loss-sensitive regularization implemented in practice? Does the optimization require computing second-order derivatives or Hessian–vector products, and if so, what is the additional time and memory cost compared to GPM or SGP?

•	What specific benefit does the LSS provide compared to simpler soft projection regularization (e.g., scaling by singular values or energy norms)? Could similar effects be achieved with first-order measures of importance?

•	Theoretical analysis (Theorem 3.1) is presented without a clear interpretation. Can the authors provide an intuitive explanation or visualization showing why the curvature-based scaling reduces projection distortion?

•	In the ablation study, what does “MTL + soft” correspond to? How is it implemented if multitask learning already assumes access to all task data simultaneously?

•	The paper should include computational complexity and runtime comparisons against prior works. How does the cost of constructing and updating the Fisher-based LSS scale with the number of tasks and layers?

•	Several relevant prior works, such as CODE-CL and other gradient-projection continual learning methods, are missing in the references. Could the authors clarify how their approach differs conceptually and algorithmically from these recent developments?

---

### Official Review · Reviewer_JboY · 2025-11-02

[review text omitted: it was posted to a different submission]

---

### Note · Authors · 2025-11-12

I have read and agree with the venue's withdrawal policy on behalf of myself and my co-authors.